# Transformations of Vascular Flora of a Medieval Settlement Site: A Case Study of a Fortified Settlement in Giecz (Wielkopolska Region, Western Poland)

**Zbigniew Celka** [1,*] **, Andrzej Brzeg** [2] **and Adam Sobczyński** [3]

1   Department of Systematic and Environmental Botany, Faculty of Biology, Adam Mickiewicz University, Poznań, Uniwersytetu Poznańskiego 6, 61-614 Poznań, Poland
2   Department of Plant Ecology and Environmental Protection, Faculty of Biology, Adam Mickiewicz University, Poznań, Uniwersytetu Poznańskiego 6, 61-614 Poznań, Poland
3   Independent Researcher, Janowo 111, 63-000 Środa Wielkopolska, Poland
*   Correspondence: zcelka@amu.edu.pl

**Abstract:** Exceptional components of the cultural landscape of Central Europe include archaeological sites, e.g., castle ruins, prehistoric or medieval fortified settlements, other settlements and burial mounds. The plants associated with them help us explain the processes of species persistence on habitat islands as well as the process of naturalization of crop species, which escape from fields or are abandoned. This study describes the flora of a medieval fortified settlement in Giecz (Wielkopolska region, western Poland), presents plant indicators of former settlements (relics of cultivation), species of high conservation value, and transformations of the vascular flora of this settlement over a few decades. Field research was conducted in 1993–1994, 1998–1999, and 2019. At the study site, 298 species of vascular plant species were recorded, and nearly 70% of them (201 species) have persisted there over the last 20 years. The flora includes seven relics of cultivation (*Artemisia absinthium*, *Leonurus cardiaca*, *Lycium barbarum*, *Malva alcea*, *Pastinaca sativa*, *Saponaria officinalis*, and *Viola odorata*), 5 species threatened with extinction in Poland and/or Wielkopolska, and 53 species of least concern (LC) according to the European red list. We have attempted to explain the floristic changes. The archaeological site in Giecz is of high conservation value, very distinct from the surrounding cultural landscape because of its specific flora, and composed of species from various habitats (e.g., dry grasslands, wooded patches, meadows, aquatic and ruderal habitats), including threatened, protected, and relic species.

**Keywords:** Central Europe; floristic study; human impact; invasive species; medieval fortified settlements; native species; plant diversity; relics of cultivation; regional flora

## 1. Introduction

In Central Europe, archaeological sites are particularly interesting components of the cultural landscape [1–3]. In Poland, for example, they include not only sites of prehistoric and medieval settlements: remnants of fortified settlements (so-called *grody*, surrounded by ramparts made of timber and earth, with a palisade and an external ditch) as well as subservient *podgrodzia*, i.e., unfortified or less fortified settlements connected with *grody*, other settlements, and abandoned and neglected castle ruins. Archaeological sites include also places of burial: cemeteries and burial mounds [4–6]. Particularly important among them are medieval sites, built between the 7th and 15th century, i.e., during the formation of statehood in many countries of Central Europe [1,7,8]. Archaeological sites scattered all over this region are significant traces of the history of its colonization by various ethnic groups, and at many of them specific flora has survived for hundreds of years [9–13]. It can be observed especially at sites that are easily distinguishable in the landscape, e.g., elevated above the neighbouring terrain, uninhabited and/or located in isolated places (such as lake islands, peninsulas, river bends, wetlands). Sites of former settlements and burial mounds

are associated with many research problems, concerning species persistence in habitats strongly transformed by human activity, on habitat islands created by archaeological sites (e.g., patches of steppe vegetation on burial mounds—kurgans—in Ukraine), and linked with naturalization of crop species, escaping from fields or abandoned, which have survived until present near and at those sites [6,11,14–16].

Floras of archaeological sites, especially sites of former settlements and burial mounds, were studied in detail in many European countries in the last few decades [16–31]. The history of research on contemporary plant cover of sites of former settlements in Central Europe started in the 18th century [32–34]. In the 20th and 21st centuries, such studies were conducted, e.g., in Czechia [11,35,36], Germany [9,11,37–41], and Poland [10,11,13,42–48]. In Mecklenburg in Germany, floristic transformations of medieval sites of Slavonic settlements have been studied for over 100 years [11,41,49–53].

The Wielkopolska region located in mid-western Poland is the place of origin of Polish statehood. The tribal state created here in the early medieval period, with a capital in Gniezno, was the seed of the Polish state with well-developed settlements, strong agriculture, and economic/trading links, it markedly influenced nature. A major role in the newly formed state was played by *grody*. They were enclosed for defensive purposes: surrounded by ring-like ramparts, made of timber and earth, with an external ditch. They were the seats of rulers, law courts, state administration, tax collectors, and armed crew. Around those sites, in the subservient *podgrodzia* connected with them and in nearby settlements, merchants as well as craftsmen and craftswomen lived and worked. Trade routes ran through them or close to them [7,54]. Fortified settlements were usually built at sites that were safer, not easily accessible, e.g., on hills, lake islands or at river bends, mentioned earlier. Their remnants, after they lost importance (damaged and abandoned), are termed *grodziska* in Polish [55]. In the Wielkopolska region alone, their number is estimated to exceed 600 [56].

In the early Middle Ages, in the developing Polish state, the most important were so-called central *grody*, each of them covering an area of several hectares, strongly fortified, with princes' palaces and churches, inhabited by rulers and their courts, state officials, and strongly armed crew [57,58]. One of several central *grody*, which was not completely destroyed (with some undamaged fragments of the stronghold) is located in Giecz in Wielkopolska. The aim of this study was (i) to describe the flora of the fortified settlement in Giecz; (ii) to draw attention to groups of species persisting in the flora of the settlement and distinguishing it, evidencing its specificity and association with medieval settlements; (iii) to present transformations of the vascular flora of this fortified settlement over 20 years; and (iv) to indicate species of high conservation value on the regional scale (Wielkopolska), national scale (Poland), and continental scale (Europe).

## 2. Materials and Methods

### 2.1. Wielkopolska as a Physiographic and Geobotanical Region

Wielkopolska (also known as Greater Poland) is a large region, covering over 38,300 km$^2$, located in the catchment area of the Warta (tributary of the Oder), which is part of the Central European Lowland. The capital of this region is the city of Poznań. The geomorphology of Wielkopolska has been shaped during the Riss glaciation and the Weichselian glaciation, i.e., the last glacial period [59].

According to the geobotanical approach (depending on classifications), Wielkopolska is part of the Baltic Division, subdivided into the Kujawy-Wielkopolska Land and Northern Edge Plateaus [60], or the Brandenburg-Wielkopolska Division, composed of the Noteć-Lubusz Land, Central Wielkopolska, Kujawy, and South Wielkopolska with Lusatia [61]. Wielkopolska is also at the eastern limit of distribution of *Fagus sylvatica* L. In the regions located further east, fewer Atlantic plant species are found, while the number of continental ones increases [60].

## 2.2. Study Site

The archaeological site in Giecz (locally known as Grodziszczko) is located 40 km east of Poznań (Figure 1). It is one of the largest, well-preserved sites of medieval settlements in Wielkopolska: 270 m long, 180 m wide (inner part 200 m long, 130 m wide), covering an area of nearly 4 hectares [62]. It can be accessed through 2 gates: from the south and from the north-east. In some places, its ramparts reach up to 9 m above the neighbouring meadows, marshes, and arable fields. It is currently situated at the edges of the Maskawa river valley, but several hundred years ago it was bounded on the eastern side by a narrow and long lake [63]. In the Middle Ages the fortified settlement was connected by a causeway and bridge with a trading settlement, located on the eastern edges of the lake [64,65].

The fortifications in Giecz were probably built in the 9th century. Stone masons started to build a small palace with a rotunda for a ruler (so-called palatium), but it has never been finished [66]. It played an important role at the beginning of Polish state formation, as one of several central *grody* [57,58]. It flourished in the 11th century, when in its northern part a stone church was built, one of the oldest in Wielkopolska. Its development was temporarily stopped in 1038 by a Bohemian prince, Bretislav I, who attacked Giecz and relocated its inhabitants to Bohemia. In the late 11th century, the rebuilt settlement became a set of prince's administration, as so-called castellany (in Polish *kasztelania*). The interior was divided by a rampart into 2 parts: the northern part included the church and castellan's headquarters, while the larger southern part, with a dense network of wooden houses, played the role of a subservient settlement. The division was maintained till the late 12th or early 13th century. Changes in trade routes, creation of towns and cities, and damage caused by fights between princes in the early 13th century, led to a decline of its importance [58,64–66]. For many ages (11th–20th centuries) a cemetery functioned there [66]. Some trees were planted in and near the cemetery, in the central part of the study site, which now constitutes a park.

Currently, the whole study site is protected as an archaeological reserve, which is part of the Museum of the First Piasts at Lednica. It includes remnants of the early medieval palatium, the Church of St. John the Baptist and Our Lady of Consolation, with the priest's house, and a new museum building constructed in 2009–2010. In the same period, south of the archaeological site, the early medieval settlement was reconstructed, with various housing types (huts and pit-houses, i.e., dugouts or semi-dugouts), a space for outdoor events, educational ventures and archaeological festivals, a building for temporary expositions, and a vegetable garden with various plants [66] (Figure 1). As an important site documenting the beginnings of Polish state, it is subject to careful restoration. Several times a year, self-sown plants are removed from the remains of the palatium, preserved as permanent ruins. The places sown with grass seeds in front of the museum are regularly mown, whereas trees and shrubs are trimmed or removed. The largest and highest rampart (9 m above the surrounding terrain) on the western side, which is visible from the car park of the museum, is mown very frequently.

## 2.3. Nomenclature and Classifications

Field research was conducted at the archaeological site and in its immediate vicinity (up to 50 m away from it) in 1993–1994, 1998–1999, and 2019. We collected and dried plant specimens and made floristic lists [67]. The dried specimens are deposited in the herbarium of the Department of Systematic and Environmental Botany of Adam Mickiewicz University in Poznań (POZ). In this study, we also used plant material collected till 1999 [68], which was verified and analysed once again.

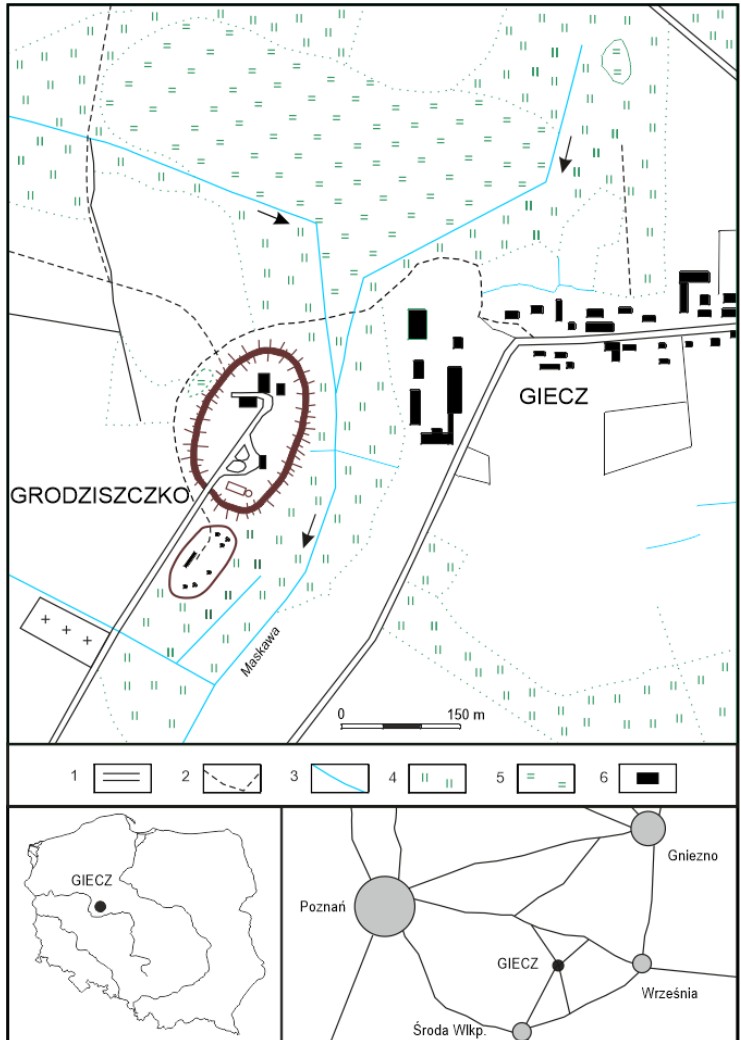

**Figure 1.** Location of the medieval fortified settlement in Giecz (after [68], modified and supplemented). 1: roads; 2: pathways; 3: watercourses; 4: meadows; 5: wetlands; 6: buildings.

Names of species and families follow Mirek et al. [69] (current scientific names of some species according to the Plants of the World Online database [70] were different, so they are given in brackets in Table S1). Classification of Raunkiaer plant life-forms follows Zarzycki et al. [71]. Geographical-historical groups are based on the concepts of Jackowiak [72], Zając and Zając [73], as well as Jackowiak et al. [74,75]. The following groups were distinguished: native species (NS) and alien ones (anthropophytes): archaeophytes (Ar: species introduced to the area or persisting there thanks to human activity before the end of the 15th century), neophytes (Ne: species introduced to the area after the discovery of America by Christopher Columbus in 1492 and permanently naturalized), and ergasiophygophytes (Ef: species escaping from cultivation) [72,76]. The applied socio-ecological classification was developed by van der Maarel [77] and Kunick [78]. It is based on grouping of syntaxa influenced by similar habitat conditions and mutual successional links. Each species was assigned to one group, according to its optimum occurrence in the phytogeographic conditions of Polish lowlands, which was reflected in its diagnostic role in the regional system of plant communities. The classes include characteristic species of all the lower syntaxa: orders, alliances, and associations, as well as diagnostic (mostly dominant) taxa for non-hierarchic communities known from the literature. In Poland the socio-ecological classification was used and developed by e.g., Jackowiak [72,79,80], Chmiel [81,82], Żukowski et al. [76], and Celka [11,44,45]. When diagnosing individual

syntaxa, we used Central European publications: Zarzycki et al. [71], Jäger et al. [83], Matuszkiewicz [84], and Ratyńska et al. [85]. The scope of anthropogenic changes in the flora of the study site was described using synanthropization indices, based on numerical relationships between geographical-historical groups. The applied indices were defined by Jackowiak [79] and Chmiel [82].

Relics of cultivation were distinguished in accordance with Bauch [37], Russow [52], and Celka [11], species threatened in Wielkopolska according to Jackowiak et al. [86], in Poland according to Kaźmierczakowa et al. [87], and in Europe according to Bilz et al. [88]. The following threat categories are used here: VU = vulnerable, NT = near-threatened, LC = least concern, DD = data-deficient [86–88]. Names of plant communities follow Matuszkiewicz [84] and Ratyńska et al. [85].

## 3. Results

Our research was focused on the vascular flora of the whole study area, not on its plant communities, but as a background for further analyses, we provide here its brief phytosociological description. Currently, the archaeological site in Giecz is bounded on the eastern side (and partly the northern and southern) by plant communities of the syntaxonomic class *Phragmitetea australis* (Figure 2). These include primarily reed canary grass beds (*Phalaridetum arundinaceae*) and numerous sedge communities (e.g., with *Carex gracilis*, *C. disticha*, *C. riparia*, *Glyceria maxima*, and *Thalictrum flavum*). On the western and partly northern sides, the site is bounded by arable fields, with typical field weeds (e.g., *Avena fatua*, *Consolida regalis*, *Papaver dubium*, and *P. rhoeas*). At the base of the rampart on the western, southern, and northern sides, a path is located, partly on a modern causeway leading to the village of Giecz, parallel to the bridge that linked the fortifications with the village in the Middle Ages.

On the ramparts of the study site (Figure 2), particularly in the western and southern parts, thermophilous grasslands of the class *Festuco-Brometea* have developed (e.g., with *Astragalus cicer*, *Campanula bononiensis*, *Phleum phleoides*, *Thalictrum minus*, and *Verbascum lychnitis*). They are frequently and precisely mown by workers responsible for maintenance of the reserve. At the northern and partly western edges of the reserve, the rampart is covered by woody plants, mostly *Robinia pseudoacacia*, *Prunus domestica*, *Rosa canina*, *Sambucus nigra*, and *Syringa vulgaris*. Part of the interior of the settlement is currently a park (Figure 2). Flower beds are composed of ornamental perennials (e.g., *Hemerocallis fulva*). The whole park is planted with numerous trees and shrubs (e.g., *Acer saccharinum*, *Aesculus hippocastanum*, *Caragana arborescens*, *Juglans regia*, *Philadelphus coronarius*, *Picea abies*, *Ribes alpinum*, *Spiraea chamaedryfolia*, *S. salicifolia*, and *Viburnum lantana*). Some of the planted species tend to escape from cultivation thanks to sexual or vegetative reproduction (e.g., *Ailanthus altissima*, *Rosa rugosa* or *Thuja occidentalis*) and were also taken into account in this study. In the north-west, at the base of the rampart, a small depression is filled with water in some periods. It is a remnant of an external ditch (moat, which surrounded the fortifications in the past), with numerous aquatic and waterside species, e.g., *Carex riparia*, *Eleocharis palustris*, *Phragmites australis*, and *Typha latifolia*).

During our field research, initiated in 1993, a total of 298 plant species were found at the archaeological site in Giecz and its immediate vicinity (Table S1). For 20 years, 201 species persisted there (67.5% of the flora). In the 21st century, 60 new species appeared (20.1%), whereas 37 (12.4%) of the species recorded in the 1990s apparently disappeared.

The spectrum of Raunkiaer plant life-forms for this archaeological site does not deviate from the Central European standard (Table 1, Figure 3). The dominant life-forms are hemicryptophytes, which with therophytes account for more than 66% of the flora (Table 1). The ratio of hemicryptophytes to therophytes is 2.2. Most of the hemicryptophytes were recorded in all the study periods (95 species, over 70%). Among the therophytes that appeared in the 21st century, alien species prevail (85%), e.g., *Eragrostis minor*, *Euphorbia helioscopia*, and *Impatiens parviflora*. The contribution of cryptophytes is also high, and they are represented primarily by geophytes and phanerophytes. Within the last mentioned

group, trees are more diverse than shrubs. The phanerophytes that appeared in the 21st century include native species associated with moist habitats (e.g., *Salix cinerea*), neophytes spreading in Poland (e.g., *Acer negundo*), and a group of naturalized cultivated plants (four species).

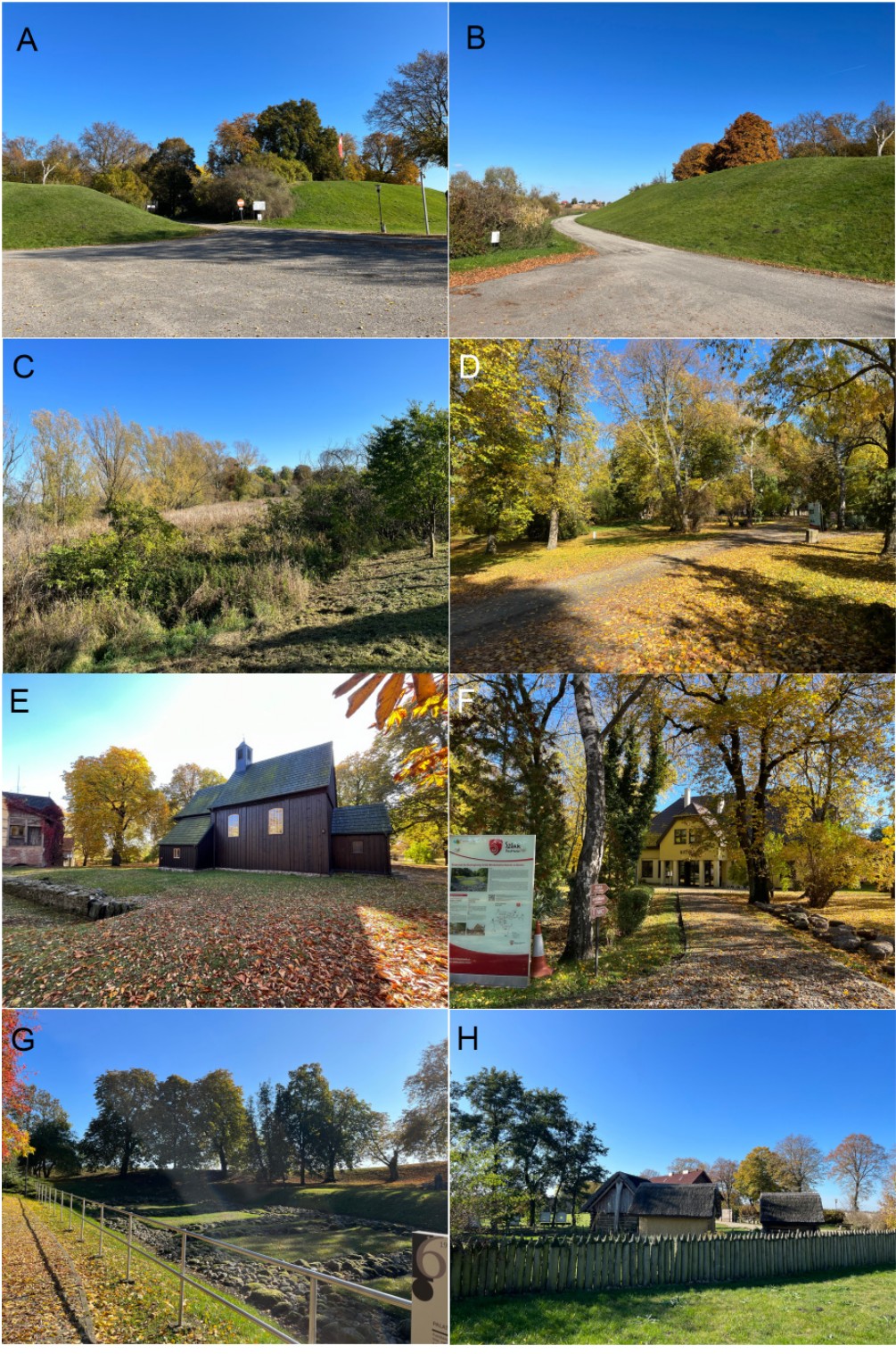

**Figure 2.** The fortified settlement in Giecz: (**A**) southern gate, (**B**) western rampart with thermophilous grassland, (**C**) marsh community on the eastern side, (**D**) interior of the study site, planted with trees, (**E**) Church of St. John the Baptist and Our Lady of Consolation, (**F**) museum building, (**G**) foundations of palatium ruins, and (**H**) educational settlement (20 October 2022, photos: A. Sobczyński).

**Table 1.** Contributions of Raunkiaer plant life-forms to the total number of species.

| Raunkiaer Plant Life-Form | Number of Species | Contribution (%) |
|---|---|---|
| Phanerophytes | 47 | 15.8 |
| Chamaephytes | 14 | 4.7 |
| Hemicryptophytes | 135 | 45.3 |
| Geophytes | 30 | 10.1 |
| Hydrophytes | 10 | 3.4 |
| Therophytes | 62 | 20.8 |

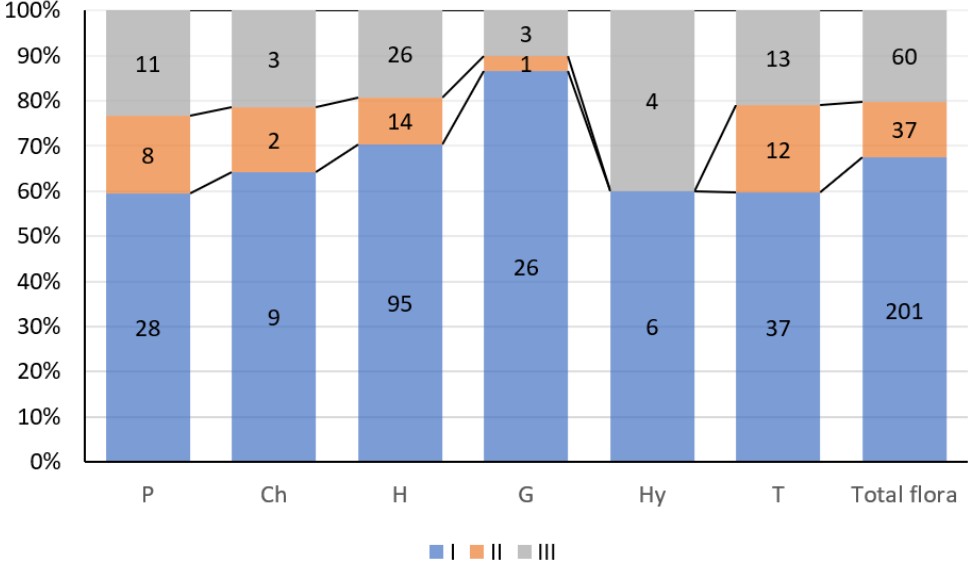

**Figure 3.** Contributions of plant life-forms to the flora of the fortified settlement in Giecz. I: species present in all study periods (since 1993). II: observed only till 1999. III: observed only in 2019. P: phanerophyte. Ch: chamaephyte. H: hemicryptophyte. G: geophyte. Hy: hydrophyte. T: therophyte.

The flora of our study site is dominated by native species, in total 203 (over 68% of the flora, Figure 4). Most of them were recorded in all the study periods (148 species, over 73%). Among the new species, which appeared in the 21st century, those associated with moist and aquatic habitats are particularly noteworthy, e.g., *Alnus glutinosa*, *Rorippa amphibia* or *Salix cinerea*. Also the earlier immigrants, so-called archaeophytes, accounted for a large proportion of the flora, as they were represented by 51 species (17.1%, Figure 4). Within this group, 32 species (63%) were recorded in all the study periods. The species that disappeared in the 21st century include e.g., *Artemisia absinthium*, *Conium maculatum*, and *Leonurus cardiaca*. The species naturalized in Poland after the 15th century (so-called neophytes) make up a small proportion of the flora (25 species, 8.4%), like the species that escaped from cultivation, so-called ergasiophygophytes (19 species, 6.4%). Within the latter group, the greatest changes in taxonomic composition are noticeable, as only five of those species were present in all study periods. They are naturalized woody plants: *Philadelphus coronarius*, *Prunus domestica* (two subspecies: subsp. *insititia* and subsp. *domestica*), *Pyrus communis*, *Syringa vulgaris*, and *Thuja occidentalis*.

Indices of anthropogenic changes of the flora (Table 2) show that the flora of the study site is strongly synanthropic (total synanthropization index $WS_c > 80\%$). Among aliens, archaeophytes prevail (total archaeophytization index $WAr_c = 18.1\%$ in 1999 and 15.3% in 2019, whereas the flora modernization index $WM < 40\%$). The contribution of diaphytes is relatively low (index of floristic fluctuations $WF = 4.2\%$ in 1999 and 5.4% in 2019). For a period of 20 years, values of most of the indices were quite stable, except for a

remarkable decrease in the contribution of archaeophytes (mentioned above) and increased contributions of neophytes (total kenophytization index $WKn_t$ from 7.1% to 9.2%, and flora modernization index WM from 28.8% to 38.1%) and ergasiophygophytes (index of floristic fluctuations WF from 4.2% to 5.4%).

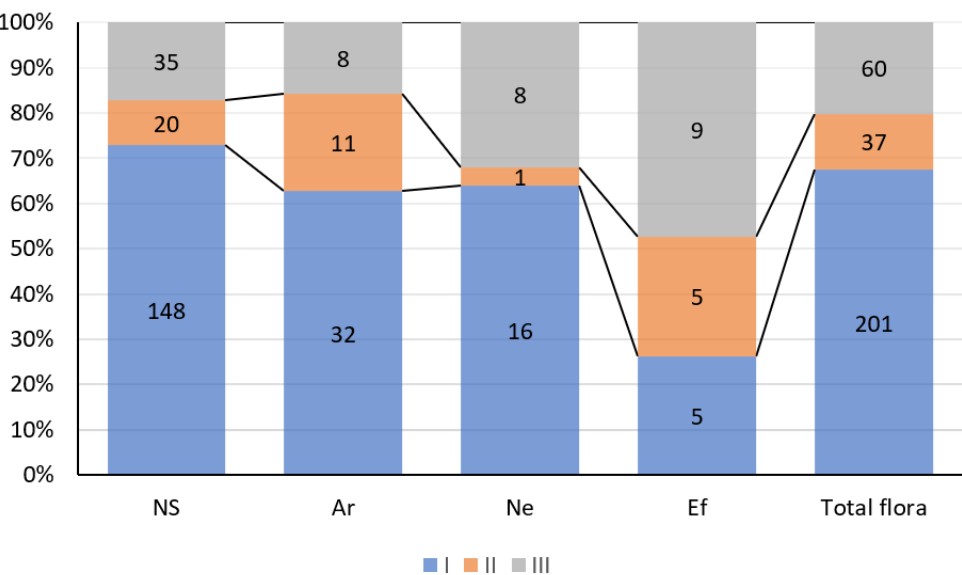

**Figure 4.** Geographical-historical composition of the flora of the fortified settlement in Giecz. I: species present in all study periods (since 1993). II: observed only till 1999. III: observed only in 2019. NS: native species. Ar: archaeophyte. Ne: neophyte. Ef: ergasiophygophyte.

**Table 2.** Indices of anthropogenic changes in the flora, defined by Jackowiak [79] and Chmiel [82].

| Indices of Anthropogenic Changes | Total Flora |
|---|---|
| Indices of flora synanthropization | |
| total ($WS_c$) | 83.9 |
| permanent ($WS_t$) | 82.8 |
| Indices of apophytization | |
| total ($WAp_c$) | 52.0 |
| permanent ($Wap_t$) | 55.6 |
| Spontaneophyte apophytization index (Wap) | 76.4 |
| Indices of flora anthropophytization | |
| total ($Wan_c$) | 31.9 |
| permanent ($Wan_t$) | 27.2 |
| Indices of flora archaeophytization | |
| total ($War_c$) | 17.1 |
| permanent ($War_t$) | 18.3 |
| Indices of flora kenophytization | |
| total ($WKn_c$) | 8.4 |
| permanent ($WKn_t$) | 9.0 |
| Flora modernization index (WM) | 33.3 |
| Index of floristic fluctuations (WF) | 6.4 |
| Flora naturalness index (WN) | 16.1 |
| Indices of flora permanence | |
| of anthropophytes ($WT_A$) | 80.0 |
| total ($WT_C$) | 93.6 |

An analysis of geographical-historical groups and life-forms in the total flora shows that no chamaephytes are neophytes and no hydrophytes are anthropophytes (Figure 5). In all Raunkiaer groups, native species prevail except for therophytes, where archaeophytes are the most numerous. It is also noteworthy that archaeophytes and neophytes are major

groups among therophytes, while ergasiophygophytes predominate among phanerophytes (Figure 5).

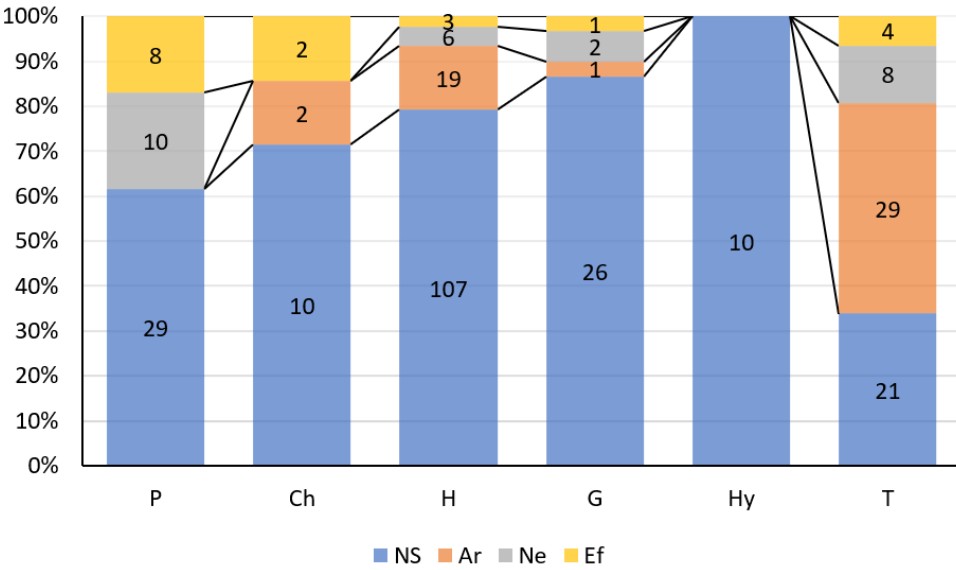

**Figure 5.** Contributions of geographical-historical groups within plant life-forms in the flora of the fortified settlement in Giecz. P: phanerophyte. Ch: chamaephyte. H: hemicryptophyte. G: geophyte. Hy: hydrophyte. T: therophyte. NS: native species. Ar: archaeophyte. Ne: neophyte. Ef: ergasiophygophyte.

The number of species in socio-ecological groups varies from 1 in the group of plants typical of fens and wet meadows (G2) to 63 in the group of plants of ruderal communities (G14) (Figure 6). No species represents the group of epilithic communities of the class *Asplenietea trichomanis* (G15). Large numbers of species represent the vegetation of meadows, pastures, and perennial trodden communities (G5), sandy and xerothermic grasslands (G6), thermophilous oak forests, mesophilous broadleaved forests, nitrophilous shrub communities (G12), and segetal communities (G13). Five of the most species-rich groups account for nearly 70% of the flora. Groups G1–G12 are dominated by native species, whereas G13, G14, and G16, by anthropophytes. However, some expansive neophytes are found in coniferous forests and acidophilous broadleaved forests (*Padus serotina*), alluvial communities (*Acer negundo*), and mesophilous broadleaved forests (*Impatiens parviflora* and *Robinia pseudoacacia*), while an archaeophyte (*Viola odorata*), in mesophilous broadleaved forests and nitrophilous shrub communities.

Central European relics of cultivation are represented by seven species at the study site in Giecz (Table 3). They are relics of medieval cultivation (*Leonurus cardiaca* and *Malva alcea*), medieval–modern cultivation (*Artemisia absinthium*, *Pastinaca sativa*, *Saponaria officinalis*, and *Viola odorata*), and relics of cultivation in the modern era (*Lycium barbarum*). Two of them were not observed in 2019 (*Artemisia absinthium* and *Leonurus cardiaca*).

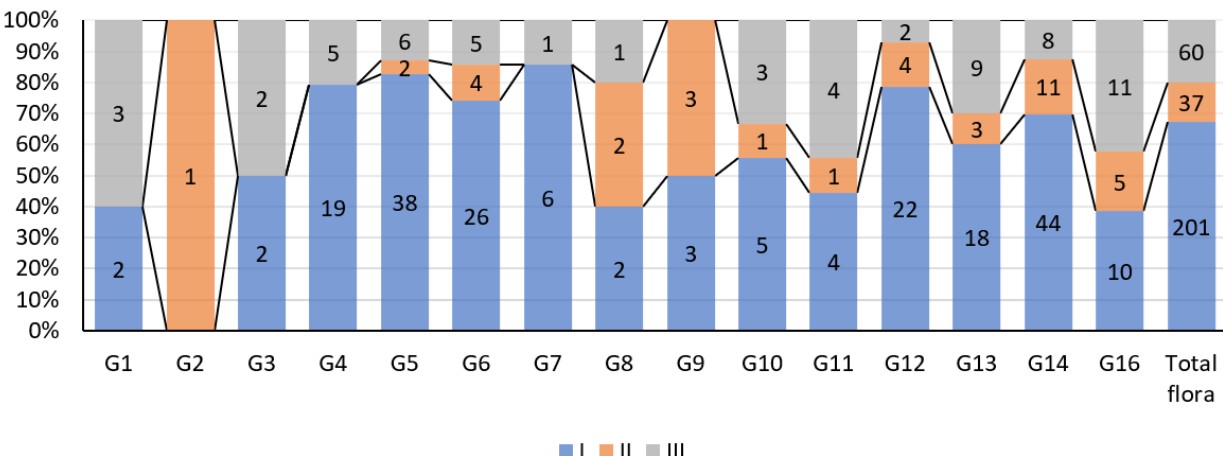

**Figure 6.** Contributions of sociological-ecological groups to the flora of the fortified settlement in Giecz. I: species present in all study periods (since 1993). II: observed only till 1999. III: observed only in 2019. G1: aquatic and flush vegetation (*Lemnetea minoris, Potametea, Utricularietea intermedio-minoris, Montio-Cardaminetea*). G2: raised bogs and bog meadows (*Scheuchzerio-Caricetea fuscae*). G3: communities of waterside therophytes (*Bidentetea tripartitae, Isoëto durieui-Juncetea bufonii*). G4: reedbeds and sedge communities (*Phragmitetea australis = Phragmito-Magno-Caricetea*). G5: meadows, pastures, and perennial trodden communities (*Molinio-Arrhenatheretea, incl. Cynosurion s.l., Potentillion anserinae*). G6: xerothermic and sandy grasslands (*Koelerio-Corynephoretea, Festuco-Brometea*). G7: thermophilous shrub communities and forest edge communities (*Trifolio-Geranietea sanguini, Rhamno-Prunetea*). G8: acid moors and forest clearings (*Calluno-Uliceta, Epilobietea angustifolii–Carici piluliferae-Epilobion angustifolii*). G9: coniferous and acidophilous broadleaved forests (*Vaccinio-Piceetea, Quercetea robori-petraeae*). G10: riparian willow carrs and willow shrub communities, riparian tall-herb communities rich in climbers (*Salicetea purpureae, Artemisietea vulgaris—Senecionion fluviatilis*). G11: alder carrs and alder-ash-elm carrs (*Alnetea glutinosae, Querco-Fagetea, = Carpino-Fagetea—Alnion incanae*). G12: thermophilous oak forests, mesophilous broadleaved forests, and nitrophilous shrub communities (*Querco-Fagetea—Carpinion betuli, Fagion sylvaticae, Tilio-Acerion pseudoplatani, Artemisietea vulgaris–Galio-Alliarion, Epilobietea angustifolii—Sambuco-Salicion capreae*). G13: segetal communities (*Stellarietea mediae—Panico-Setarion, Veronico-Euphorbion, Scleranthion annui, Caucalidion lappulae*). G14: ruderal communities (*Stellarietea mediae—Malvion neglectae, Salsolion ruthenicae, Sisymbrion, Artemisietea vulgaris—Arction lappae, Convolvulo arvensis-Agropyrion repentis, Onopordion acanthi s.l., Polygono-Poetea annuae*). G16: species of undetermined phytosociological affiliation (mostly ergasiophygophytes and some neophytes).

*Artemisia absinthium* was found in Giecz in ruderal habitats and at the edges of a thermophilous grassland on the rampart of the study site. The species was infrequent, forming several patches. It was used in many periods, for various purposes (see Table 3). *Leonurus cardiaca* was also infrequent, in ruderal habitats of the interior of the study site, on the rampart with thermophilous grassland, along the path surrounding the rampart, and in the garden near the museum building as a weed in cultivation. It was used mostly as a medicinal plant (Table 3), but currently it is primarily a ruderal plant. *Lycium barbarum* was used in the modern era mostly as an ornamental and for this purpose it was planted in the study area. In Giecz it grows in ruderal patches and thickets on the rampart, with *Syringa vulgaris* and *Robinia pseudoacacia*, probably as result of dispersal from the park, where it was cultivated. *Malva alcea* is one of the most common relics of medieval cultivation for various purposes at archaeological sites of Central Europe (see Table 3). In the fortified settlement in Giecz it is very rare on the thermophilous grassland on the rampart but frequent in the immediate neighbourhood of the study site: chiefly along the river Maskawa, on roadsides and roadside ditches, and near the bridge, where it spread from places of its cultivation linked with the archaeological site. *Pastinaca sativa* is rare in the study area, associated with ruderal habitats in its interior, on the grassland on the rampart, and along the paths

surrounding the fortifications. It was cultivated mostly as a vegetable. *Saponaria officinalis* grows in Giecz mostly on the grassland, in ruderal habitats, and rarely at the edges of ruderalized shrub communities. Its presence in the study area can be linked not only with its cultivation for medicinal purposes but also (or even mostly?) as an ornamental, funeral species planted in the cemetery, which was located in the interior of the fortified settlement. In such places it is currently found in many localities in Wielkopolska and in Poland. *Viola odorata* grows primarily in the internal park, on the grassland on the rampart, and in ruderalized shrub communities, or—less frequently—in ruderal habitats. Its presence in the study area can be linked not only with its medieval cultivation for medicinal purposes, but also with modern cultivation as an ornamental plant, very much like *Saponaria officinalis*, linked with the cemetery.

**Table 3.** Relics of cultivation in the fortified settlement in Giecz after [11], modified and supplemented.

| Species | Cultivation Period | | | Major Use | | | | | Found in Archaeological Deposits | Frequency at Archaeological Sites of Central Europe |
|---|---|---|---|---|---|---|---|---|---|---|
| | **Pr** | **Md** | **Me** | **M** | **F** | **P** | **D** | **O** | | |
| *Artemisia absinthium* | + | + | + | + | + | + | + | + | + | ** |
| *Leonurus cardiaca* | - | + | + | + | - | + | + | + | + | ** |
| *Lycium barbarum* | - | - | + | + | - | - | + | + | - | * |
| *Malva alcea* | + | + | + | + | + | + | + | + | + | *** |
| *Pastinaca sativa* | + | + | + | + | + | - | - | + | + | ** |
| *Saponaria officinalis* | +? | + | + | + | - | + | + | + | + | * |
| *Viola odorata* | + | + | + | + | - | - | + | + | + | ** |

Pr = prehistoric; Md = medieval; Me = modern era; M = medicinal; F = food and seasonings; P = pigments; D = decorative; O = others; * = widely distributed; ** = frequent; *** = very frequent.

During field research, 5 species threatened in Poland and/or Wielkopolska were recorded. The most interesting among them is *Campanula bononiensis* (category VU in Wielkopolska and NT in Poland), recorded since the early 1990s, in all study periods. It grows on the thermophilous grassland on the western part of the rampart, in spite of frequent mowing. In favourable conditions it flowers and bears fruit. Since 2004 it is also on the list of protected species in Poland. *Calamagrostis stricta* (category VU in Wielkopolska and NT in Poland) grew in a marsh community in the remnants of the ditch on the western side and on the river Maskawa, but in 2019 within the study area it was absent. *Euphorbia exigua* (rare in Poland, category NT) grew on ruins of the palatium. It is a small therophyte, which copes well if it does not need to compete with other species, frequently removed by workers caring from palatium ruins. However, in 2019 it was not observed there. *Populus nigra* (category LC in Wielkopolska) grows on the banks of the river Maskawa. In 2019 in a ruderal habitat several individuals of *Veronica polita* were found (in Poland category DD = Data Deficient, i.e., a taxon whose threat category cannot be precisely determined because of information shortage). The recorded species from the European red list include 53 classified as LC (Least Concern, which according to IUCN criteria cannot be classified as CR, EN or VU, but still they deserve attention and protection) as well as *Malus sylvestris* classified as DD (see Table S1).

## 4. Discussion

Transformation of the natural environment under the influence of human activity started when humans appeared on the Earth. On the one hand, the changes lead to the extinction/decline of some species [89–91], but on the other hand, to the spread of invasive aliens [92–94]. Material traces of human activity in the past are preserved at archaeological sites all over Europe. They often form a true mosaic of habitats. Within one archaeological site we can find a thermophilous grassland, a ditch filled with water, and a wooded habitat. Part of the site can be cultivated for crops, ornamentals or mown as a lawn/meadow [9,11,16,26,39,44]. Sites of medieval settlements and burial mounds located in rural or urban landscapes are sometimes the only places that increase local

biodiversity thanks to the plant communities that developed on them, e.g., wooded patches or grasslands among fields or in towns/cities [44,95,96]. Such sites are refuges of rare, endangered or vulnerable, protected, and endemic species of plants, animals and fungi [11,16,21,29,31,44,97–101]. Among archaeological sites, those situated in locations that are not easily accessible are particularly noteworthy (e.g., at river bends, on peninsulas, and on lake islands). They not only are refuges for plants, but their floras change much slower than floras of sites of former settlements that are subject to strong human impact in agricultural or urban landscapes [11,37,41,49,50,53]. The medieval fortified settlement in Giecz is relatively big, covering an area of nearly 4 hectares [62], with a high habitat diversity. It was built at the edges of a lake that dried out. On its ramparts, thermophilous grasslands have developed, as well as spontaneously formed thickets with *Robinia pseudoacacia*, *Syringa vulgaris*, *Sambucus nigra*, and *Rosa canina*. Part of the interior of the settlement is currently a park, with flower beds and other ornamentals. The study site includes also ruderal sites and small vegetable patches. The fortifications are bounded by marsh plant communities of the class *Phragmitetea australis*. In some places and periods, depending on water level in the river Maskawa, these communities are flooded, especially the remnants of the external ditch protecting the settlement from the north-western side. The fortifications are located close to cultivated fields, a parking lot, and roads (surfaced or not). The large size of the study site and the diversity of habitats resulted in development of an exceptionally rich flora here, as compared to other sites of medieval or earlier fortified settlements in Poland [11,44].

Changes in floras of medieval settlement sites are mostly caused by human activity [9,39,40,102] and less often by natural processes, such as ecological succession or floods [16]. Nearly 70% of species composition at the study site remained unchanged, in spite of its substantial transformations in the last 2 decades. For nearly 30 years, 201 species have persisted there. In the 21st century, 60 new ones have appeared, while 37 species are apparently missing (see Figure 3). As a result of sanitary and husbandry practices in the course of site maintenance as well as construction of the museum building and educational settlement, some species observed at the site in the 20th century were probably completely removed. This applies to species growing near or in the place of the new buildings, e.g., *Dryopteris filix-mas*, *Malus sylvestris*, *Populus tremula*, *Rubus idaeus* or *Sorbus aucuparia* [31,103]. Changes in the flora of wetland, waterside, and aquatic species are probably due to rising water level in the river Maskawa but also to human interference: so-called "cleaning" of the river channel, mowing of waterside vegetation, and removal of trees and shrubs [104,105]. About a dozen native species disappeared or appeared in such habitats. The new species are typical of aquatic habitats (*Potamogeton crispus*, *Spirodela polyrhiza*, *Stuckenia pectinata*), wetlands and waterside vegetation (*Berula erecta*, *Rorippa amphibia*, *Sium latifolium*, *Valeriana officinalis*) as well as alluvial habitats (*Alnus glutinosa*, *Humulus lupulus*, *Salix cinerea*, *S. fragilis*, *Solanum dulcamara*). Plants of thermophilous (xerothermic) grasslands at the archaeological site in Giecz are subject to strong human impact, consisting in mowing many times every year, so that sexual reproduction is impossible or difficult for many species. Despite this, the most valuable grassland species recorded in the study area—*Campanula bononiensis*—has survived there for many years, forming a very small local population, very much like most of the other thermophilous species.

The changes in the flora of the study site are partly consistent with the changes observed in the flora of Wielkopolska [74,75,106], Poland [69,107], or—more broadly—Europe and the world [89,91,108]. This applies to disappearance of rare and threatened native species [87,88,109] and archaeophytes [69,109], but simultaneous spread of neophytes and appearance of ergasiophygophytes [92–94]. As compared to research conducted in the 20th century, at this site still 31 archaeophytes persist (see Table S1). We did not find 11 species, but 8 new ones were recorded. Presumably some archaeophytes declined or were removed due to changes in land use in the study area [110]. This applies to several species that are not red-listed but are declining in Poland, e.g., *Fumaria officinalis* subsp. *officinalis*, *Leonurus cardiaca*, *Solanum nigrum* or *Urtica urens* [107,111], but also to archaeo-

phytes that are frequent or even invasive in some parts of Poland, e.g., *Conium maculatum*, *Descurainia sophia*, *Digitaria ischaemum*, *Lepidium ruderale* or *Sisymbrium officinale* [107,111]. Instead, some common archaeophytes appeared: *Euphorbia helioscopia*, *Papaver argemone*, *Senecio vulgaris*, *Sinapis arvensis*, *Viola arvensis* [111,112], and some rarer ones: *Consolida regalis*, *Chamomilla recutita*, *Veronica polita* [111]. Probably some of the changes in species composition of archaeophytes result from accidental dispersal of therophytes, originating from the immediate neighbourhood of the study site. The neophyte flora of the archaeological site in Giecz is growing continuously. Throughout the 3 decades of research, 16 neophyte species have persisted (64%). Most of them are invasive woody species: *Ailanthus altissima*, *Lycium barbarum*, *Padus serotina*, *Robinia pseudoacacia*, *Rosa rugosa*; or invasive therophytes: *Amaranthus retroflexus*, *Conyza canadensis*, *Galinsoga ciliata*, *Galinsoga parviflora* [31,113–116]. Changes in the neophyte flora of the study site concern 36% of the total number, but in 2019 only one of them was missing (*Datura stramonium*). Instead, eight new ones appeared. They include common invasive species, spreading for many years in Poland and globally: *Acer negundo*, *Impatiens parviflora*, *Lolium multiflorum*, *Sisymbrium loeselii* [24,74,112–114]; or spreading only recently: *Eragrostis minor* [117–121]; or those escaping from cultivation and naturalized quickly: *Mentha × niliacea*, *Symphoricarpos albus* [114]. Changes in the group of ergasiophygophytes are particularly numerous, which results from their specificity [75,113,122,123]. In the late 20th century they were represented by 10 species, but only 5 of them were recorded in 2019, namely cultivated and naturalized woody species: *Philadelphus coronarius*, *Prunus domestica*, *Pyrus communis*, *Syringa vulgaris*, *Thuja occidentalis*. The missing species are mostly annual, such as *Secale cereale* and *Triticum aestivum*, ornamental *Myosotis sylvatica*, and fruit-bearing *Fragaria × ananassa*. Instead, nine new species appeared (mostly garden escapes near the educational settlement: *Alcea rosea*, *Cnicus benedictus*, *Lavandula angustifolia*, *Linum usitatissimum*, and *Ruta graveolens*.

Relics of cultivation, grown by various ethnic groups and tribes in the past for various purposes, now are no longer cultivated or cultivated only sporadically [11,37,44,51,52,124,125]. We find them in floras of many countries and on many continents [108,126–132]. They were grown as medicinal and decorative plants, sources of seasonings, food, pigments, forage, as well as plants cultivated for magic, cosmetic, and many other purposes (see Table 3). Relics of cultivation in Central Europe are currently divided into three subgroups: relics of medieval cultivation, medieval-modern ones, and relics of cultivation in the modern era [11]. Large-scale research on those species was undertaken in Central Europe [11,37,51,53,133–136]. The relics include 22 species [11], and in Giecz 7 of them were found, which is the highest number of species of this group at a single archaeological site in Wielkopolska except for those on the peninsula in Kórnik-Bnin and Pierska Island [11]. Relics of medieval cultivation were represented by *Leonurus cardiaca* and *Malva alcea*, medieval-modern ones by *Artemisia absinthium*, *Pastinaca sativa*, *Saponaria officinalis* and *Viola odorata*, while relics of cultivation in the modern era by *Lycium barbarum*. In 2019, we recorded only five of them (*Artemisia absinthium* and *Leonurus cardiaca* were absent). Medieval relics of cultivation survive in places where they were formerly cultivated and rarely have a tendency to spread along transportation routes (roads and railway tracks). In Mecklenburg, local populations of the relic species (e.g., *Allium scorodoprasum*, *Malva alcea*, *Origanum vulgare*), which were observed in the 19th century, persist at sites of medieval Slavic settlements [11,137,138]. Research conducted on lake islands with archaeological sites in former Neustrelitz district in Mecklenburg (north-eastern Germany) after 50 years showed that most of the relics recorded there survived, but the number of their local populations declined [41,49,50,53]. Relics of cultivation at archaeological sites sometimes form small populations, composed of only several individuals, which can persist there for many years thanks to vegetative propagation or a seed bank in the soil, but they may be difficult to find. To determine if a given species completely disappeared from a site repeated floristic research is necessary in several growing seasons [11,44].

The flora of the fortified settlement in Giecz, which was particularly important for Polish history over 1000 years ago, is one of the richest in Central Europe. Despite the

changes observed in the last 3 decades, its main part—composed of more than 200 species—remains unchanged. The 37 that were not recorded after 20 years cannot be classified as extinct there, as their persistence in the changing conditions of the site in Giecz may be revealed in the future. Its flora is distinguished from the surrounding habitats, as species of various habitat types (e.g., dry grassland, woodlot, meadow, aquatic or ruderal habitats) coexist there, including species threatened with extinction in Wielkopolska, Poland, and Europe, protected in Poland, and indicators of former settlements, i.e., relics of cultivation.

**Supplementary Materials:** The following supporting information can be downloaded at: https://www.mdpi.com/article/10.3390/d15010035/s1, Table S1: List of species recorded in the medieval fortified settlement in Giecz (Wielkopolska, western Poland).

**Author Contributions:** Conceptualization, Z.C. and A.B.; methodology, Z.C. and A.B.; software, Z.C.; validation, Z.C. and A.B.; formal analysis, Z.C. and A.S.; investigation, Z.C., A.B. and A.S.; resources, Z.C., A.B. and A.S.; data curation, Z.C.; writing—original draft preparation, Z.C.; writing—review and editing, Z.C. and A.B.; visualization, Z.C.; supervision, Z.C.; project administration, Z.C.; funding acquisition, Z.C. and A.B. All authors have read and agreed to the published version of the manuscript.

**Funding:** This research funded by the Faculty of Biology, Adam Mickiewicz University, Poznań, Poland.

**Institutional Review Board Statement:** Not applicable.

**Data Availability Statement:** Not applicable.

**Conflicts of Interest:** The authors declare no conflict of interest.

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
