# Peer review of "Transformations of Vascular Flora of a Medieval Settlement Site: A Case Study of a Fortified Settlement in Giecz (Wielkopolska Region, Western Poland)"

_diversity, doi:10.3390/d15010035_

Round 1
Reviewer 1 Report
The authors examine an interesting topic (Transformations of vascular flora of a medieval settlement site: 2 a case study of a fortified settlement in Giecz (Wielkopolska re- 3 gion, western Poland). The authors made appropriate amount of field work. But sadly, the lack of precision methodology required in scientific publication is lacking in the method section. Almost entire Ms revolve around the habitat types but in the methodology section author fails to explain how many habitat types are present the study area. Many interesting data but its interpretation is not convenient many times and the manuscript contains a series of sloppy formulations that are not acceptable in a peer reviewed scientific article. However, the worthy data in the manuscript deserve publication and thus I give some advice with which the authors could make a new manuscript which could be published for example in this journal.

Author Response
Dear Reviewer,
Thank you very much for your favourable opinion about our manuscript, all the remarks, and comments. Below we have listed our answers. The whole text was additionally checked by a science editor in respect of language.
Our study concerned changes in the vascular flora of the study area, not changes in vegetation (plant communities) or in floras of individual habitats. Part of the information about vascular plants in plant communities (habitats) at and near the archaeological site (mostly the first 2 paragraphs of Results and e.g. in lines 182, 189, 308) is only a background for the further analyses and discussion. This is now explained at the beginning of the section Results. Phytosociological nomenclature has been verified and updated, although it does not play a major role in this study.
Phytosociological nomenclature and approach to syntaxa, as well as their floristic characteristics, follow recent monographs by Polish authors, mainly Matuszkiewicz (2008), supplemented or corrected by Ratyńska et al. (2010), both added to the list of references. In the latter source, valid names of units were verified in detail according to the International Code of Phytosociological Nomenclature (ICPN). In some cases, synonymic names of classes have been given after foreign (mainly Czech and German) authors.
Matuszkiewicz W. 2008. Przewodnik do oznaczania zbiorowisk roślinnych Polski. 540 pp. Wydawnictwo Naukowe PWN, Warszawa.
Ratyńska H., Wojterska M., Brzeg A., Kołacz M. 2010. Multimedialna encyklopedia zbiorowisk roślinnych Polski, ver. 1.1. (CD). NFOSiGW, UKW, IETI. Bydgoszcz.
The section Results starts from a general description of the plant cover of the archaeological site and its immediate vicinity, to present a broader background of the study area and facilitate understanding of the text and e.g. proper interpretation of Figure 2. That is why we propose to leave the first 2 paragraphs of Results unchanged, with a brief explanation added at the beginning. The suggested move of information from line 214 can be placed after these 2 paragraphs.
In the section Materials and Methods we have added short information about the sources of names of syntaxa. This study, however, concerns transformations of the flora of vascular plants of the whole study area. Our field research and plant material collection were not focused on transformations within habitats or plant communities, so we cannot make any reliable analysis of this aspect. Nevertheless, this may become an objective of our further research.
Line 150
I suppose you meant Raunkiaer plant life-forms. They are now listed in Table 1.
Line 152
Reference added.
Lines 169 and 214
Explained above.
Line 173
Explained above.
Line 174
Explained above.
Line 182
We analysed the flora of the whole study area, not of individual habitat types or ramparts on the western and southern side. More explanations above.
Line 189
Explained above.
Line 199
We have added a new table with information about species number in each Raunkiaer plant life-form. Numbering of successive tables has been corrected.
Line 214
We have placed this paragraph after the first 2 paragraphs of Results.
Line 308
Explained above.
All the syntaxa have been verified/corrected in the whole manuscript according to the sources listed below (and already mentioned above), which have been added to the list of references. In the section Materials and Methods we have added short information about the sources of names of syntaxa.
Matuszkiewicz W. 2008. Przewodnik do oznaczania zbiorowisk roślinnych Polski. 540 pp. Wyd. Nauk. PWN, Warszawa.
Ratyńska H., Wojterska M., Brzeg A., Kołacz M. 2010. Multimedialna encyklopedia zbiorowisk roślinnych Polski, ver. 1.1. (CD). NFOSiGW, UKW, IETI. Bydgoszcz.
Line 333
The paragraph in Materials and Methods provides more details now.
We are grateful for all remarks and suggestions. We hope that our responses are satisfactory.
Kind regards,
Authors
Reviewer 2 Report
The manuscript is higly valid and well done. The part about phytosociological components of the flora can be improved and updated, even though it doesn't have a primary role in the paper. Here some generic observation. More detailed observations as comments in the manuscript.
In table S1 species labelled with II are wrongly labelled with III and vice versa. It can be easily seen both through counting numbers of every category (total number, native species, archeophytes, etc.) and with single species.
Ergasiophygophytes and ergasiophytes are used as synonims? If yes as it seems, I suggest to use only one of them.
The reference to phytosociological units have to be updated.
The way used to group phytosociological units (fig. 6) is questionable abut some groups, particularly G7, G8, G10, G11 and G12. No papers are cited about studies on the vegetation of the area and on the dynamical links among plant communities. If that way of grouping is original, it is needed to better describe those links. Moreover, nothing is said in the discussion about changes in species linked to meadows and xerothermic and sandy grasslands (groups G5 and G6 in fig. 6) that are not irrelevant .

Author Response
Dear Reviewer,
Thank you very much for your favourable opinion about our manuscript, all remarks, and comments. Below we have listed our answers.
Line 13
Corrected. The other settlements were not fortified.
Line 15
Corrected as “naturalization”.
Line 145
Corrected.
Line 155
The term „Socio-ecological” is necessary, see below for explanations (line 157).
Line 157
Phytosociological nomenclature and approach to syntaxa, as well as their floristic characteristics, follow recent monographs by Polish authors, mainly Matuszkiewicz (2008), supplemented or corrected by Ratyńska et al. (2010), both added to the list of references. In the latter source, valid names of units were verified in detail according to the International Code of Phytosociological Nomenclature (ICPN). In some cases, synonymic names of classes are given after foreign (mainly Czech and German) authors.
Matuszkiewicz W. 2008. Przewodnik do oznaczania zbiorowisk roślinnych Polski. 540 pp. Wydawnictwo Naukowe PWN, Warszawa.
Ratyńska H., Wojterska M., Brzeg A., Kołacz M. 2010. Multimedialna encyklopedia zbiorowisk roślinnych Polski, ver. 1.1. (CD). NFOSiGW, UKW, IETI. Bydgoszcz.
Additional explanations concern the socio-ecological classification.
Plant species were divided into socio-ecological groups on the basis of the classification introduced by Kunick (1974) and refers to the concept of van den Maarel (1971), who distinguished 15 socio-ecological groups by linking phytosociological groups with similar characteristics and various successional links. Each species was assigned to one group, according to its optimum of occurrence in the phytogeographic conditions of Polish lowlands, which was reflected in its diagnostic role in the regional system of plant communities. The classes include characteristic species of all the lower syntaxa: orders, alliances, and associations as well as diagnostic (mostly dominant) taxa for non-hierarchic communities known from the literature. In Poland the socio-ecological classification was applied and developed by e.g. Jackowiak (1990, 1993, 1998), Chmiel (1993, 2006), Å»ukowski et al. (1995), and Celka (1999, 2004, 2011). When diagnosing individual syntaxa, we used Central European publications: Zarzycki et al. (2002), Jäger et al. (2005), Matuszkiewicz (2008), and RatyÅ„ska et al. (2010). In this manuscript we use the classification based on the above sources, whereas changes resulting from the use of Mucina et al. (2016) would require in some cases far-reaching modifications, which were not objectives of our research and thus are outside the scope of our study. This is now explained in detail in the section Materials and Methods and we hope that the explanation will be sufficient for the reviewer and readers.
Line 166
Explained above (line 157)
Line 170
Yes, the current name is C. acuta. In our manuscript we use the names of species according to Mirek et al. (2020), prepared especially for Poland. We have added international synonyms in brackets in Table S1 and information in the text.
Line 210, Fig. 3
We have corrected Figure 3 as suggested.
Line 224
Corrected.
Line 242
Thanks for your comment.
Line 255
Corrected.
Line 257, Fig. 5
We have corrected Figure 5 as suggested.
Line 262
The term “socio-ecological groups” is necessary. See our explanations for line 157.
Line 275, Fig. 6
We have corrected Figure 6 as suggested.
Line 289
Corrected.
Line 413
A relevant sentence has been added. Thank you for the suggestion!
Line 424
Corrected.
Answers to general remarks:
We have corrected the mistakes in Table S1. Thank you for your help.
We have also revised the explanations under Table S1.
“Ergasiophygophytes” and “ergasiophytes” were used as synonyms here. For consistency, we now use in the whole manuscript the term “ergasiophygophytes”.
Thank you for all remarks and suggestions. We hope that our responses are satisfactory.
Kind regards,
Authors
Round 2
Reviewer 1 Report
Agree with the revision made by the author. Accept Ms in present form